# Environmental Life Cycle Analysis of Açaí (*Euterpe oleracea*) Powders Obtained via Two Drying Methods

Natalia Salgado-Aristizabal [1], Tatiana Agudelo-Patiño [1], Sebastian Ospina-Corral [2], Ignacio Álvarez-Lanzarote [2] and Carlos Eduardo Orrego [3,*]

1 Instituto de Biotecnología y Agroindustria, Universidad Nacional de Colombia, Manizales 170003, Colombia
2 Departamento de Producción Animal y Ciencia de los Alimentos, Tecnología de los Alimentos, Facultad de Veterinaria, Instituto Agroalimentario de Aragón-IA2-(Universidad de Zaragoza-CITA), 50013 Zaragoza, Spain
3 Departamento de Física y Química, Instituto de Biotecnología y Agroindustria, Universidad Nacional de Colombia, Manizales 170003, Colombia
* Correspondence: ceorregoa@unal.edu.co

**Abstract:** Açaí is a fruit native to Brazil that is found in Colombia, and it is recognized for containing more than 90 compounds with anticancer, anti-inflammatory, and other biological activities. In this study, a cradle-to-gate life cycle analysis (LCA) was conducted for the production of açaí powder, following the methodology outlined in the ISO 14040 standard. The investigation focused on examining the impact of utilizing or not utilizing the residues generated during the pulp extraction step as fertilizers. Four scenarios were analyzed and compared: (i) production of açaí powder via vacuum drying, (ii) via spray drying, and via the same two types of drying but using residues from the pulping operation as fertilizer (Scenarios 3 and 4). It was found that to produce 1 kg of açaí in a crop cycle, 1.17 kg of $CO_2$ eq is produced. The drying stage in Scenarios 1 and 2 generated 8.04 and 7.93 kg of $CO_2$ eq, respectively. Similarly, when solid waste is used as fertilizer, $CO_2$ emissions barely increased for Scenarios 3 and 4, respectively. To the authors' knowledge, this is the first carbon footprint study of the production of açaí powder whit these scenarios.

**Keywords:** *Euterpe oleracea*; vacuum drying; spray drying; environmental life cycle analysis; waste management; açaí crop





## 1. Introduction

The açaí (*Euterpe oleracea*) is a palm species (family *Arecaceae*) that produces fruit with a high demand in Brazil and other countries. Açaí pulp contains proteins, lipids, and fibers and is highly energetic [1]. It also has bioactive compounds, such as anthocyanins and carotenoids, with therapeutic, anti-inflammatory, and anticancer properties [2]. The food and pharmaceutical industries have shown interest in this fruit [3]. The pulp can be used as an ingredient in various products, such as candies, yogurts, or energy drinks, among others [4].

Several studies have reported on the processing and characterization of açaí pulp powder. Pavan et al. [5] investigated the water sorption isotherms and conducted thermal analysis of açaí powder using three drying methods. Lucas et al. [6] evaluated the changes in the physical properties and the bio-compounds of the açaí pulp dried using different methods, while de Almeida Magalhães et al. [7] conducted a similar study on freeze-dried pulp. Other authors have determined the optimal operating conditions for the spray-drying process [1] as well as the use of carrier agents to improve its performance [8].

Colombia is the top exporter of exotic fruits in South America and accounts for 15 percent of the world's exotic fruit sales [9]. Some of the over 400 types of fruit in the country can help poor and remote areas that have been affected by violence. For example, açaí, which grows in hard-to-reach jungle areas, is one of these fruits. As it is not easy to sell

the fruit due to its location, processing fruit that is not consumed fresh becomes crucial for extending its shelf life. Dehydration is one of the simplest methods for this. However, it is important to study how much the process of drying fruit pulp impacts the environment before implementing it in açaí farms [10]. This is because drying can consume a significant amount of energy in a process [11]. For instance, pasteurization requires between 17% and 26% of the energy required to make milk [12], while drying consumes 51% of the total energy required for making milk powder [13].

Açaí pulp constitutes only 20% of the fruit, with the rest often going to waste, leading to pollution [14]. Some studies have used açaí waste to extract antioxidants [8] and to produce biochar [15], compost [16], and other products [17].

Various research works have explored the environmental impacts of different drying technologies [10,18] as well as the utilization of waste from the drying process to minimize harm [19]. However, to the authors' knowledge, no study has been published specifically on açaí drying. Moreover, tropical forests are rich in renewable resources and medicinal plants. These green and renewable sources can help prevent rainforest destruction by stimulating demand for sustainable products. These products not only support the livelihoods of local communities but also discourage deforestation for timber extraction. Therefore, researchers should employ innovative or traditional technologies to produce sustainable and healthful raw food derivatives that efficiently utilize forest resources.

This study conducts a cradle-to-gate life cycle analysis (LCA) of açaí powder production in accordance with the ISO 14040 standard [20]. Additionally, it examines the potential utilization of pulping waste as fertilizer. Four scenarios are evaluated: (i) production of açaí powder via vacuum drying, (ii) production of açaí powder via spray drying, (iii) production of açaí powder via vacuum drying with the incorporation of pulping waste as fertilizer, and (iv) production of açaí powder via spray drying with the incorporation of pulping waste as fertilizer.

The LCA in this paper is of an explorative nature as large-scale commercial açaí farms are not yet established in Colombia. However, some small-scale pilots have been initiated in the rainforest areas for fresh fruit production. The aim of the LCA in this study is to examine future dried açaí production systems in the cultivation regions and identify potential optimization strategies. This information could be beneficial for stakeholders involved in the açaí supply chain in Colombia's jungle regions.

Tropical forests are abundant in nuts, fruits, oil-producing plants, and medicinal plants. These renewable resources can prevent rainforest destruction by creating demand for sustainable products that support the livelihoods of local communities and discourage deforestation. This study aims to examine the environmental impact of açaí powder production and identify potential directions for optimization. Its significance lies in its ability to provide insights into reducing energy consumption during drying processes, utilizing processing residues as fertilizer, and scaling production to achieve greater efficiency. By assessing the carbon footprint of different drying methods and the inclusion or exclusion of processing wastes, this study can inform future açaí dry powder production systems in cultivation regions and contribute to the development of more sustainable practices.

## 2. Materials and Methods

### 2.1. Raw Material and Experimental Setup

The açaí fruit were obtained from growers in Quibdó-Choco and were pulped 48 hours after being harvested. In order to obtain the açaí pulp, the fruit were washed (0.46 L kg$^{-1}$) and disinfected with running water and a solution of water with hypochlorite at 0.5% vol. in a 4:3 ratio, respectively (solid to liquid). Next, the fruit were briefly immersed in water at 50 °C to soften them and to facilitate the manual extraction of their pulp, which was packed in plastic bags and stored at −20 °C for later use and characterization. Both the fruit pulp and residues (seeds and slurry) were weighed to establish the material balance of the operation. Before the drying experiment, maltodextrin (20 DE) was added until a mix of 30° Brix was obtained. To obtain the açaí powders, the pulp–maltodextrin mixture was dried

in laboratory-scale vacuum dryer and spray dryer. Vacuum drying was performed with a custom-designed equipment featuring a vacuum pump (ISE SAS), a pressure measurement and control system (Pfeiffer Vacuum GmbH, Aßlar, Germany), electrical resistance, and a condenser cooled by a refrigerated bath/circulator. Spray drying was performed with a mini Buchi model 191 spray dryer (Büchi Laboratoriums Technik, Flawil, Switzerland).

The operating conditions of the dryers are summarized in Table 1. The drying yield was calculated according to Equation (1).

$$\text{yield}(\%\text{wt}) = \frac{\text{Recovered solids(dry base)}}{\text{Feed solids(dry base)}} \times 100 \tag{1}$$

**Table 1.** Operating conditions to obtain açaí powder.

| | **Vacuum Drying** | | | **Spray Drying** | |
| Parameter | Value | Unit | Parameter | Value | Unit |
| --- | --- | --- | --- | --- | --- |
| Shelf heating temperature | 55 | °C | Air inlet temperature | 120 | °C |
| Condenser temperature | −40 | °C | Air outlet temperature | 60 | °C |
| Pressure | 17 | mbar | Feed flow | 72 | mL/min |
| Drying time/batch [1] | 6 | h | Drying time/batch | 1.5 | h |

[1] Batch: 150 mL açaí + maltodextrin mix.

Pulp–maltodextrin samples were weighed, then dried using the drying equipment. After cooling, the dried açaí powders were reweighed, and their moisture contents were determined via drying at 105 °C in a moisture balance MOC-120H (Shimadzu Corporation, Kyoto, Japan). The powder samples were packed in high-barrier plastic bags at refrigerated temperature (4 °C) for further analysis.

## 2.2. LCA of Dried Açaí Powder

The life cycle assessment (LCA) study has been conducted in accordance with the ISO 14040 standard, following an attributional approach. The LCA encompasses all life cycle phases, including goal and scope, life cycle inventory (LCI), life cycle impact assessment (LCIA), and life cycle interpretation (ISO 14040, 2006). The assumptions and data are detailed in the following sections, beginning with the definition of the goal and scope in Section 2.2.1. The functional unit and system boundaries are described in Sections 2.2.2 and 2.2.3, respectively. The inventory data and the assumptions for each of the four açaí powder production routes are detailed in Section 2.2.4. Section 3 provides an overview of the impact assessment method used in this study.

### 2.2.1. Goal and Scope Definition

The goal of this study was to determine and compare the environmental performance of four different routes for valorizing the açaí pulp into açaí–maltodextrin powder. As previously mentioned in the introduction, the goal was to identify strategies for optimizing future commercial açaí powder production drying methods from an environmental standpoint. As illustrated in Figure 1, the scope of the study is from cradle to gate, comprising the following stages (see Figure 1): seed production, including nursery, germination, and irrigation; plant production, including digging, substrate addition, harvest, packaging, and transport; pulp extraction, including cleaning, disinfection, softening, and pulping; Scenario 1: pulp conditioning, vacuum drying, and packaging; Scenario 2: pulp conditioning, spray drying, and packaging; Scenario 3: pulp conditioning, vacuum drying, and packaging with the reuse of pulping wastes; and Scenario 4: pulp conditioning, spray drying, and packaging with the reuse of pulping wastes.

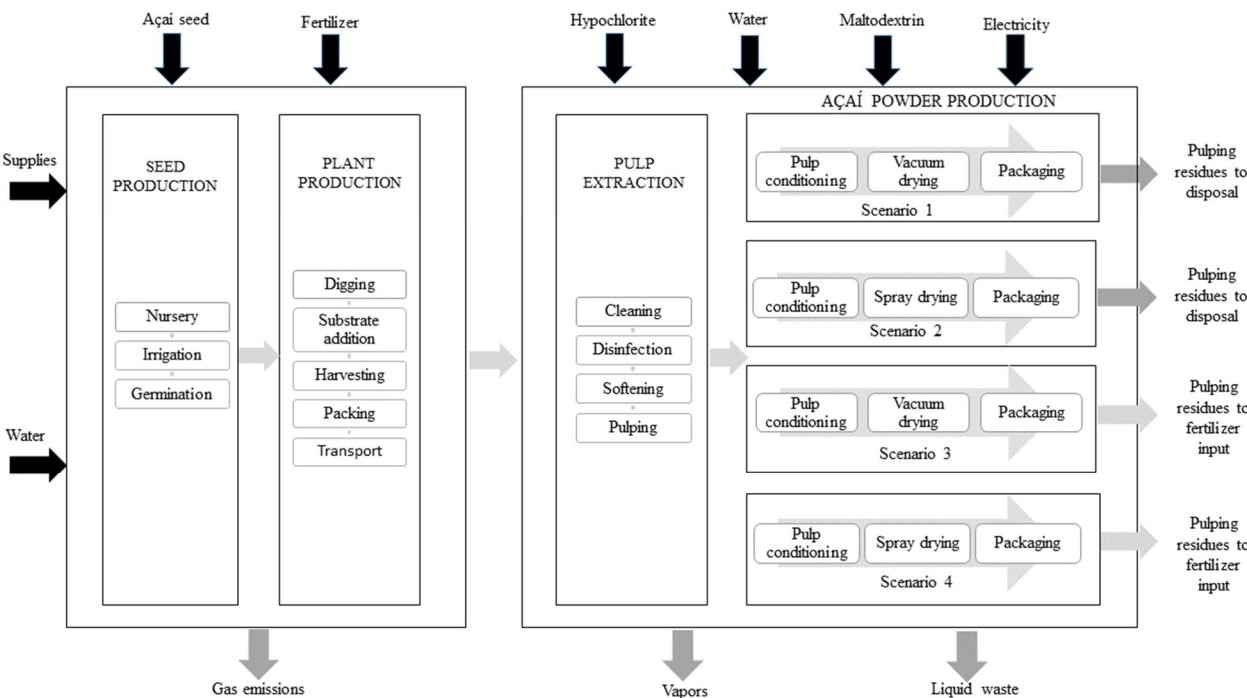

**Figure 1.** Stages and scenarios for açaí dry powder production.

2.2.2. Functional Unit

The function of the transformation system in this LCA was to produce dried açaí powder (with a moisture content: 3.5 ± 0.3% w.b.) that is stable and suitable for further processing as a component of formulated food products. The functional unit was defined as "1 kg of dehydrated açaí powder (with a 3.5% moisture content) that can be commercialized as a food ingredient". This definition helps to identify how the environmental impacts of its production change if the powder is manufactured via four different routes. The selection was based on a mass-functional unit [21–23]. Downstream processing of the dried açaí powder into commercial food products is not included in the present study.

2.2.3. System Boundaries

The system boundaries are presented in Figure 1. The açaí cultivation stage is divided into seed production and seedling production at the crop site. Fruit powder production comprises pulp extraction and the four scenarios analyzed to produce the fruit powder via vacuum drying and spray drying: two that considered only the drying methods (Scenario 1 and Scenario 2) and two that use the same two drying methods but with residues from the pulping operation as fertilizer (Scenarios 3 and 4).

2.2.4. Inventory Data Selection and Description

For the assessment, sources of primary and secondary information were used. The foreground life cycle inventory for the processes included in the system boundaries was based on empirical data from farming and measurements performed at the laboratory scale, while the background life cycle inventory database was based on the Ecoinvent database version 3. The information on the stage of cultivation was provided by growers in the region of Chocó-Colombia. Mass, energy balance, and açaí powder yield data were obtained from laboratory experimentation.

The inputs necessary for the agronomic phase, including seed and plant production as depicted in Figure 1, for one hectare up to the initial harvest of the crop are detailed in Table 2.

**Table 2.** Inputs required in the agronomic stage for 1 hectare until the first harvest of the Açaí based on practices in Chocó-Colombia.

| Stage | Activity | Time (Month) | Compounds Inlet Input | Amount | Unit | Equipment/Supplies Inlet Input | Amount | Unit |
|---|---|---|---|---|---|---|---|---|
| Nursery | Nursery construction | 1 | | - | - | Wooden supports | 144 | kg |
| | | | - | - | | Polyethylene | 5.04 | kg |
| Germinator | Bed construction | 1.5 | | | | Wood bed | 288 | kg |
| | Substrate | | Land | 288 | kg | | | |
| | Bags | | | - | - | Polyethylene | 11.81 | kg |
| Bags transplant | Substrate | | Land | 2.81 | kg | | | |
| | | | Organic fertilizer | 14.41 | kg | | | |
| | Irrigation | | Water | 236.25 | m$^3$ | | | |
| Hollowed | Substrate | 0.1 | Organic fertilizer | 1875 | kg | | | |
| Harvest | Bag | 0.1 | | | | Polyethylene | 0.022 | kg |
| | Basket | | | | | Polyethylene | 20.00 | kg |
| Packaging | Packing | 0.002 | | | | Ethylene–Vinyl–Alcohol (EVAL) | 0.06 | kg |

- Açaí cultivation (see Figure 1).

Seed production: The germination efficiency of seeds during the nursery phase is 95%. Consequently, to achieve a density of 625 trees per hectare, an additional 31 seeds are required (656 seeds in total). Generally, the nursery is located within the same crop area, eliminating the need for transportation. The seeds are manually extracted from the fruit and placed in a wooden germinator with a soil substrate. It takes about 45 days for the seeds to complete the germination process. After germination, the seeds are transplanted into bags filled with a mixture of soil and organic fertilizer at a 1:1 mass ratio to promote seedling growth. The seedlings remain in these bags for 6 months and receive 2 liters of water daily. Additionally, the resources used for nursery construction, such as wood and shade polyethylene covers, are also considered.

Plant production: Açaí, being a perennial crop, is typically harvested for 25 years, which includes 3 years of establishment and 22 years of maintenance. It yields an average of 15 tons of fruit per hectare per annum, with each tree producing 16 to 32 kg of fruit per year (average of 24 kg) [24]. The agricultural production subsystem, which includes soil management, fertilization, sowing, pest and disease management, and harvesting, does not include transportation from the farm to the dryer, as the drying system is suggested to be located on the same farm.

Due to the climatic conditions in the zone, açaí farms practice rainfed cultivation, and no weed control measures are provided. The land layout is square-shaped, with a spacing of 4 m between trees. Manual drilling is performed with a shovel, and 3 kg of organic matter is added to each hole. During plant growth, the remaining stems are cut using axes to promote the growth of the main palm tree. The time between planting and the first harvest is approximately 3 years. Manual harvesting is performed by trained personnel who climb the palm trees and cut the bunches with knives. The bunches are placed on the ground over a canvas to prevent possible contamination. The fruits are then removed and selected from the cluster, placed in plastic baskets, and taken to the collection center located within the same crop sector, eliminating the need for vehicle transportation. At

the collection center, another selection process is carried out to obtain the best fruits for transformation.

- Açaí powder production:

Pulp extraction at the Dehydration Plant: To obtain açaí pulp at the dehydration plant, açaí fruits are washed with water at a ratio of 0.46 liters of water per kilogram of fruit. Following this, the fruits are disinfected using water containing a hypochlorite concentration of 0.1% *v/v*, using the same washing ratio. The excess water from these processes is disposed of as waste.

The açaí fruits are then softened by immersing them in water at a temperature of 50 °C, using a ratio of 0.38 liters of water per kilogram of fruit. Subsequently, the açaí pulp is obtained through mechanical abrasion and carefully packed into 1 kg bags. In Scenarios 1 and 2, the seeds and slurry are considered as residues, while in Scenario 3, they were treated as a by-product. Table 3 provides an overview of the mass inputs and outputs for Scenarios 1 and 2. The mass balances for Scenario 3 remain the same except for the fact that the waste from the pulping process is utilized as a fertilizer.

**Table 3.** Mass inputs and outputs of Scenarios 1 and 2 for the production of 1 kg of açaí powder.

| Scenario | Stage | Activity | Input | Value | Unit | Output | Value | Unit |
|---|---|---|---|---|---|---|---|---|
| Scenario 1 | Pulping | Cleaning | Fruit | 4.11 | kg | | | |
| | | | Water | 1.90 | kg | | | |
| | | Disinfection | Water | 1.90 | kg | | | |
| | | | Sodium hypochlorite | 0.002 | kg | | | |
| | | Softening | Water | 2.15 | kg | | | |
| | | | | | | Liquid wastes | 1.88 | kg |
| | | | | | | Solid wastes * | 1.63 | kg |
| | Raw material conditioning | Maltodextrin addition | Maltodextrin | 1.39 | kg | | | |
| | Drying | Vacuum drying | | | | Water steam | 1.24 | kg |
| | | | | | | Açaí powder | 1.00 | kg |
| | | | | | | Açaí powder moisture | 0.03 | kg |
| Scenario 2 | Pulping | Cleaning | Fruit | 5.54 | kg | | | |
| | | | Water | 2.56 | kg | | | |
| | | Disinfection | Water | 2.56 | kg | | | |
| | | | Sodium hypochlorite | 0.003 | kg | | | |
| | | Softening | Water | 2.90 | kg | | | |
| | | | | | | Liquid wastes | 5.12 | kg |
| | | | | | | Solid wastes * | 4.43 | kg |
| | Raw material conditioning | Maltodextrin addition | Maltodextrin | 1.87 | kg | | | |
| | Drying | Spray drying | | | | Water steam | 1.25 | kg |
| | | | | | | Açaí powder | 1.00 | kg |
| | | | | | | Açaí powder moisture | 0.03 | kg |

* For Scenarios 3 and 4, pulping solid waste is used as fertilizer.

Vacuum drying (Scenario 1): To enhance drying efficiency and ensure the stability of the açaí powder, maltodextrin was incorporated into the pulp. The maltodextrin—pulp mixture, with a concentration of 30° Brix, is subjected to vacuum drying according to the conditions outlined in Table 1. The resulting encapsulated powder is then vacuum packed to maintain its quality. As previously mentioned, the solid remnants obtained from the

pulping process, namely the slurry and seeds, are designated as waste and not utilized further in production.

Spray drying (Scenario 2): Açaí powder encapsulation was achieved using the same pulp conditioning with dehydration via spray drying.

Scenarios 3 and 4: These scenarios are the same as Scenarios 1 and 2 with the difference that the pulp residues are used as fertilizers.

The energy for the drying and packaging processes was sourced from hydroelectric power plants, representing a renewable and sustainable energy source. The energy consumption of the drying equipment varied depending on the type of dryer used. For the vacuum dryer, the energy consumption included the control system, heating plate, cooler, and vacuum pump. Conversely, the spray dryer's energy consumption comprised the air heat exchanger, air suction equipment, and feed pump. These components play crucial roles in the drying process.

To ensure accurate determination of actual energy consumption, measurements were meticulously taken throughout the entire drying process using a Multimeter Fluke 117 True RMS Original (Fluke, Indonesia). This method guarantees precise and reliable data collection. Table 4 provides comprehensive details on the energy consumption of each scenario, including both the energy requirements for the drying processes and the manufacturing of packaging supplies.

**Table 4.** Energy consumption of the scenarios and for the supplies for packaging.

| Scenario | Activity | Input | Value | Unit |
|---|---|---|---|---|
| Scenario 1 | Vacuum drying | Heating plate | 0.88 | kWh |
| | | Vacuum pump + Cooler + Control system | 14.62 | kWh |
| | Packaging | Ethylene–Vinyl–Alcohol (EVAL) | 0.018 | kg |
| | | Energy consumption | 0.0095 | kWh |
| Scenario 2 | Spray Drying | Heat exchanger | 0.54 | kWh |
| | | Aspirator motor and peristaltic pump | 4.84 | kWh |
| | Packaging | Ethylene–Vinyl–Alcohol (EVAL) | 0.018 | kg |
| | | Energy consumption | 0.0095 | kWh |

### 2.2.5. Data Analyses

The life cycle assessment (LCA) was conducted using SimaPro V. 9.3.0.0 software (Pre-Sustainability, Netherlands). The EcoInvent V3.8 database was utilized for the analysis. The ReCiPe Midpoint 1 (H-Hierarchism) method was applied.

### 2.3. Sensibility Analyses

This study entailed the calculation of energy consumption, taking into account a range of key variables during the drying process to experimentally obtain commercial-quality açaí powder. In the case of spray drying, the drying temperatures were set at 120 °C, 160 °C, and 200 °C. For vacuum drying, three different chamber operation pressures were evaluated: 17 mbar, 40 mbar, and 60 mbar. The remaining drying operating conditions were held constant. The mass and energy balances served as input data for the simulation of drying process using Superpro Designer version 10 software (Intelligen, Inc., Scotch Plains, NJ, USA).

### 3. Results and Discussion

### 3.1. Drying Process

The initial moisture content and soluble solid content of the açaí pulp were 89.59% (by weight) and 30° Brix, respectively. In the context of açaí dry powder production (system 2 in Figure 1), the results obtained for the drying yield and final moisture content (on a wet basis) of the açaí encapsulates are as follows:

Scenario 1:

- Drying yield: 59.23 ± 2.08%
- Final moisture content: 3.06 ± 0.14%[a]

Scenario 2:

- Drying yield: 44.15% ± 3.20%
- Final moisture content: 2.76% ± 0.23%[a]

The values provided represent the average results obtained from three pilot plant tests. The indicated ranges represent the observed variability in the data. No significant difference was found in the final moisture at a 95% confidence level.

The drying yield, expressed as a percentage, illustrates the proportion of the original weight of the pulp that remains as dried powder after the drying process. This yield, as a measure of the efficiency of the drying process, reflects how well the process retains the initial solid content of the samples. It indicates the fraction of the initial solid content that remains in the dried powder once the drying process is completed.

The final moisture content represents the residual moisture present in the dried açaí powder encapsulates. It was observed that the powder yield in Scenario 1 was higher than that in Scenario 2. This discrepancy may be due to the tendency of fruit powders to adhere to the walls of the drying chamber during spray drying, owing to their stickiness. This is a primary reason for the use of drying aids or encapsulants when spray drying materials with high levels of low molecular weight sugars, such as fruits. The inclusion of wall material or encapsulant increases the glass transition temperature of the pulp, reducing its stickiness. Conversely, vacuum drying minimizes material loss during the removal of the dried product, since, in this equipment, the sample remains static on the drying tray during the dehydration process. This characteristic helps mitigate the issue of adherence typically encountered in spray drying.

The yield obtained in spray drying for açaí powder production in this study is similar to the findings reported by Tonon, Brabet, and Hubinger [1]. They achieved a yield of 45.74% for açaí powder using spray drying, with a final moisture content of 1.97% by weight. In their study, maltodextrin was used as the wall material at a concentration of 26% by weight.

Additionally, Du et al. [25] obtained a yield of 41.6% by weight for persimmon powder using spray drying, with a maltodextrin concentration of 25% by weight. These comparisons highlight the consistency of the drying yields achieved in spray drying processes for different fruit powders, emphasizing the role of maltodextrin or other wall materials in facilitating the drying process and improving powder yield.

*3.2. Environmental Results*

Figure 2 illustrates the contribution of the main impact categories to the total environmental impact of the açaí cultivation (System 1 in Figure 1) associated with the production of 1 kg of fresh açaí fruit. The figure provides an overview of the relative importance of different environmental factors.

The categories that contributed the most to the total environmental impact of açaí cultivation stage for obtaining 1 kg of açaí were climate change (CC), terrestrial acidification (TA), ionizing radiation (IR), freshwater ecotoxicity (FET), water depletion (WD), fossil depletion (FD), human toxicity (HT), agricultural land occupation (ALO), and metal depletion (MD). Among these categories, fertilization was identified as the stage that generated the greatest contribution in most of the evaluated categories. This can be attributed to the use of organic fertilizer, which may introduce microbiological contamination into the environment. As a result, the categories of water depletion, human toxicity, and land occupation were the ones most impacted.

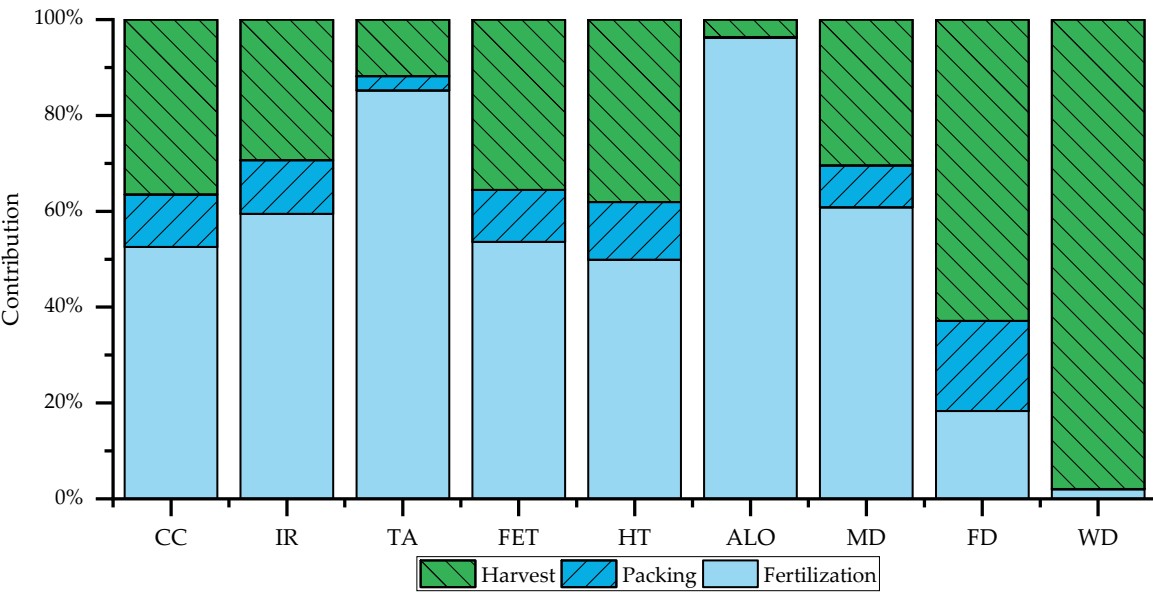

**Figure 2.** Contribution of the main categories to the total environmental impact of açaí crop. (Climate change (CC), ionizing radiation (IR), terrestrial acidification (TA), freshwater ecotoxicity (FET), human toxicity (HT), agricultural land occupation (ALO), metal depletion (MD), fossil depletion (FD), and water depletion (WD)).

These findings are important to consider when assessing and developing strategies to minimize the environmental impact of açaí crop production. Implementing sustainable fertilization practices and ensuring proper management of organic fertilizers can help mitigate the identified impacts on water resources, human health, and land use [26].

Figure 3 presents the life cycle impact evaluation of producing 1 kg of açaí powder for the analyzed scenarios using the ReCiPe Midpoint method (hierarchical version H) in SimaPro V 9.3.0.0 software. The figure is divided into four panels, each representing a different scenario: (a) Scenario 1—vacuum drying without waste use, (b) Scenario 2—spray drying without waste use, (c) Scenario 3—vacuum drying with waste use, and (d) Scenario 4—spray drying without waste use.

The figure illustrates the contribution of the main systems or stages to the total environmental impact of açaí powder production. The most representative impact categories of the ReCiPe Midpoint method are considered.

The categories that showed the greatest impact across the scenarios include climate change (CC), terrestrial acidification (TA), human toxicity (HT), terrestrial ecotoxicity (TET), freshwater ecotoxicity (FET), ionizing radiation (IR), agricultural land occupation (ALO), water depletion (WD), metal depletion (MD), and fossil depletion (FD). These categories encompass a range of environmental concerns, such as greenhouse gas emissions, pollution, resource depletion, and land use.

In terms of the specific stages evaluated, the impacts of pulp conditioning, electricity consumed for the drying process and packaging processes, packaging, and wastes were grouped together. This suggests that these stages contribute significantly to the overall environmental impact of açaí powder production.

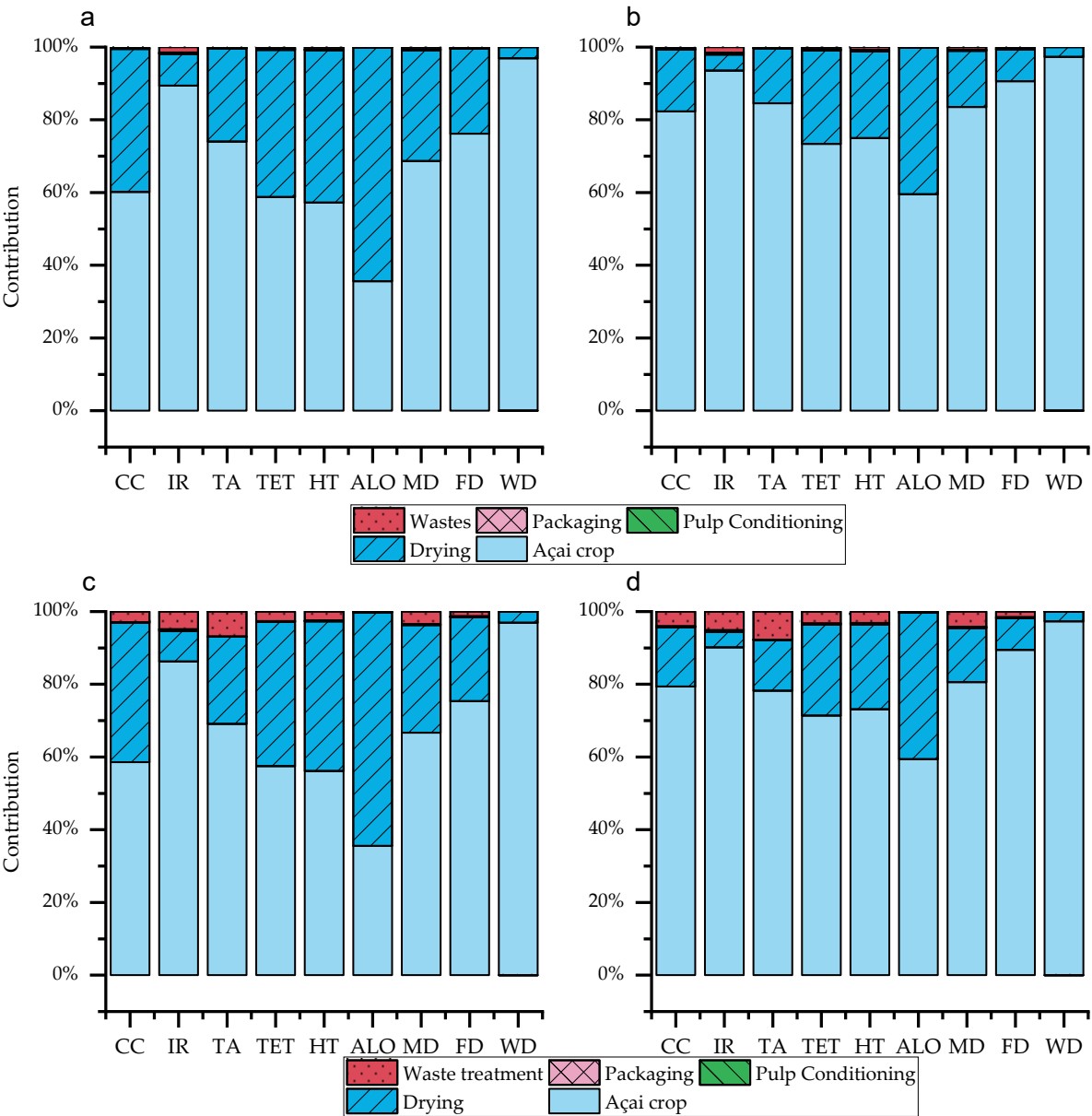

**Figure 3.** Contribution of the main categories to the total environmental impact for producing one kg of açaí powder under proposed scenarios (**a**) Scenario 1; (**b**) Scenario 2; (**c**) Scenario 3, vacuum drying; (**d**) Scenario 4, spray drying.

It is worth noting that while açaí cultivation had the highest contribution to all categories (4.81 and 6.52 kg $CO_2$ eq for vacuum drying and spray drying respectively), when evaluating the drying processes alone without considering this stage, the electrical energy consumed in the drying process emerged as the highest contributor to the environmental impact. On the other hand, packaging and pulp conditioning have little to no impact in comparison with the other stages of the life cycle, not being perceptible in the figure. Table 4 reports detailed data regarding the energy consumption associated with each drying method, providing further insight into the environmental implications of these processes.

This information provides insights into the key environmental hotspots and allows for the identification of areas where interventions or improvements can be targeted to reduce the overall environmental impact of açaí powder production.

The carbon footprint is an important standard for environmental impact assessments. There are no published works reporting the carbon footprint of açaí production. In this work, the contribution to the climate change category was 1.17 kg of $CO_2$ eq to produce

the required quantity to produce 1 kg of açaí fruit in a crop cycle. This value is lower than those published for orange crop (1.42 kg of $CO_2$ eq/kg of orange [19], coconut fruit (2.4–6, kg of $CO_2$ eq/kg of coconut), and peanuts (1.8–3 kg of $CO_2$ eq/kg of peanuts) [27]. For the cultivation of lulo, pineapple, and avocado, the carbon footprint ranges between 1.7 to 2.4 kg of $CO_2$ eq per kilogram of fruit [28]. On the other hand, Barrera-Ramírez et al. [29] reported 3.18, 0.042, and 1.3 kg of $CO_2$ eq for 1 kg of coffee, sugar cane, and cocoa, respectively, grown in Colombia. These values affirm the low environmental impact of this crop from the Chocó region due to it being a non-commercial crop under the management of cultural practices, which benefits the carbon footprint transformation process.

In the cultivation of açaí, the addition of organic fertilizer contributes 53% in the CC category, but in other categories, it had a greater impact, such as in ALO, IR, and FET with 96%, 60%, and 53%, respectively (See Figure 2).

Indeed, one of the concerns with organic fertilizer is the potential for microbiological contamination in water resources and soil if not properly managed. Pathogenic microorganisms present in organic fertilizers can pose risks of toxicity to humans and animals [30]. However, despite this drawback, the use of organic fertilizers is generally considered more beneficial than chemical fertilizers due to their lower environmental impact. Studies, such as the one by Kitamura et al. [31], have shown that organic fertilizers can be used without reducing crop yields and can even lead to a reduction in carbon footprint compared to chemical fertilizers. The study reported reduction in carbon footprint reduction from 36.2% to 16.5% when using organic fertilizers instead of chemical fertilizers.

Regarding the carbon footprint of açaí powder production, this study is the first to assess it, considering different drying methods and the inclusion or exclusion of processing wastes. In Scenarios 1 and 2, the production of 1 kg of açaí powder was found to generate 8.04 and 7.93 kg of $CO_2$ eq, respectively, as shown in Table 5. It is worth noting that Scenario 1 had a higher carbon footprint due to its higher energy consumption.

**Table 5.** Total contribution of the main categories to the environmental impact of the scenarios.

| Category | Unit | Sc 1 | Sc 2 | Sc 3 | Sc 4 |
|---|---|---|---|---|---|
| CC | kg $CO_2$ eq | 8.04 | 7.93 | 8.26 | 8.22 |
| TA | kg $SO_2$ eq | 0.07 | 0.08 | 0.07 | 0.09 |
| HT | kg 1.4-DCB eq | 4.16 | 4.28 | 4.24 | 4.39 |
| TET | kg 1.4-DCB eq | 15.24 | 16.45 | 15.57 | 16.90 |
| IR | kBq U235 eq | 0.14 | 0.18 | 0.15 | 0.19 |
| ALO | $m^2 \times$ yr | 6.00 | 4.84 | 6.01 | 4.85 |
| WD | $m^3$ | 10.25 | 13.76 | 10.25 | 13.76 |
| MD | kg Fe e | 0.01 | 0.02 | 0.01 | 0.02 |
| FD | kg oil eq | 2.45 | 2.78 | 2.48 | 2.81 |

The energy consumption in the form of electricity contributed 31.3% to the climate change category in Scenario 1, while in Scenario 2, it contributed 17%. This difference can be attributed to the higher energy consumption of vacuum drying, which requires the operation of additional equipment such as a cooler and a vacuum pump.

Comparative studies on the life cycle assessment (LCA) of food-drying processes have been conducted by various authors. For example, De Marco et al. [18] reported a carbon footprint of 8.02 kg $CO_2$ eq for a gate-to-gate apple powder package (3 kg) production using the spray drying process. Prosapio et al. [32] studied a cradle-to-grave freeze-drying of strawberries and reported a carbon footprint of 1.28 kg $CO_2$ eq for the production of 0.45 kg of strawberry powder.

In this study, the electrical energy used for drying was sourced mainly from hydroelectric power plants, which have a lower environmental impact (3.92 kg $CO_2$ eq/kWh) compared to other energy sources such as wind and nuclear power [33]. However, it is important to consider that there are other associated environmental loads specific to each

plant type, size, local climate, site characteristics, and other factors, which may cause local environmental impacts.

The values represent the total contribution of each category in the specified units for Scenarios 1, 2, 3, and 4. The categories include climate change (CC), terrestrial acidification (TA), human toxicity (HT), terrestrial ecotoxicity (TET), ionizing radiation (IR), agricultural land occupation (ALO), water depletion (WD), metal depletion (MD), and fossil depletion (FD).

It can be observed that the scenarios in general share more or less the same environmental impact. This is associated principally with the scale from which the LCI data were obtained (laboratory scale). However, Scenario 2 generally has lower values across the board. The difference between Scenario 1 and 2 is expected to be greater as the scale of production rises. The specific impacts and their magnitudes vary depending on the category, indicating the environmental implications of each scenario.

The comparisons between Scenarios 1 and 3 as well as between Scenarios 2 and 4 demonstrate that producing fertilizers from the residues generated during production barely increases the overall pollution generated by the process, indicating the potential environmental and economic benefits of utilizing processing residues in food transformation processes.

Other studies have also highlighted the positive impact of incorporating waste utilization into food production. For example, in cassava starch production, the anaerobic digestion of its residues resulted in a 28% reduction in $CO_2$ equivalent generation [34]. Similarly, in citrus processing, including pectin extraction, the overall $CO_2$ generation was reduced [35].

These findings support the notion that incorporating waste utilization strategies can contribute to more sustainable and environmentally friendly production processes when adding value to natural products. By effectively managing and utilizing waste streams, it is possible to minimize environmental impacts and promote sustainability in production projects.

### 3.3. Sensitivity Analysis

In this study, a sensitivity analysis was carried out to find the impact of variations in the main operating conditions of the drying processes. This analysis considered the minimum to maximum range of the key variables in the drying operation, which allowed for the experimental production of commercial-quality açaí powder. Energy consumption for spray drying was evaluated by adjusting the drying temperature between 120 °C, 160 °C, and 200 °C, while for vacuum drying, three chamber operation pressures—17 mbar, 40 mbar, and 60 mbar—were evaluated. The other drying operating conditions were not modified.

According to Figure 4, the spray drying Scenarios (2 and 4) exhibit a minimal escalation in their environmental footprint, and no significant difference was observed between the scenarios and temperatures. This is because the energy required to heat the air from 120 °C to 140 °C or 200 °C is relatively insignificant compared to the total energy consumed during the drying process (0.54, 0.77, and 1.2 kWh, respectively).

In the vacuum drying Scenarios (1 and 3), the differences in vacuum pressure have no significative differences. It has been observed that the energy consumption by the pump at a pressure of 17 mbar surpasses that at 40 mbar or 60 mbar (14.62, 13.74, and 13.36 kWh, respectively). Therefore, it can be confidently asserted that in terms of environmental impact, the system is not responsive to variations in operating pressure, at least at the laboratory scale.

Upon comparing vacuum drying and spray drying at the laboratory scale, no significant differences were observed between the environmental footprint of the two methods. However, it is expected that as the operations are scaled up, spray drying will prove to be more environmentally friendly compared to vacuum drying.

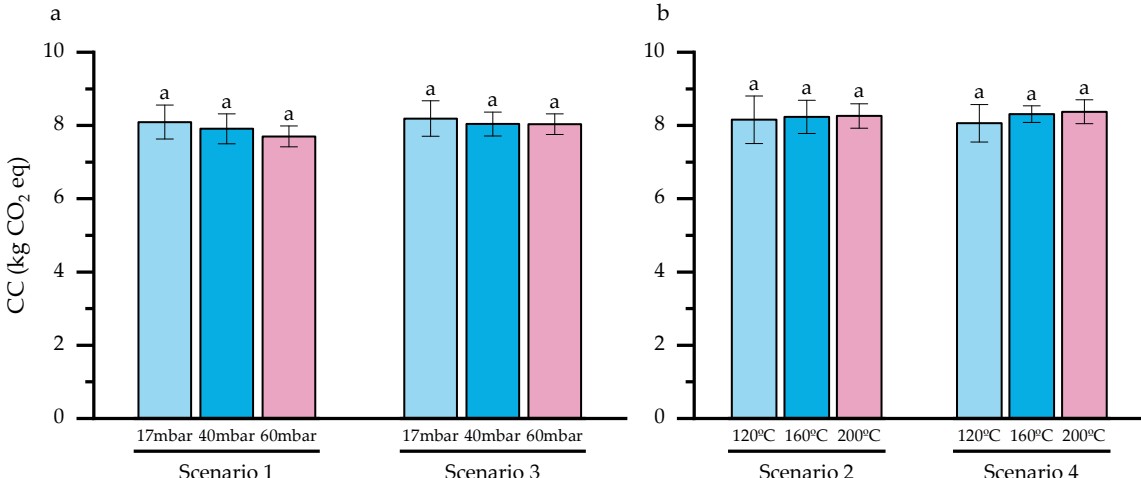

**Figure 4.** Sensitivity analyses (**a**) Scenarios 1 and 3; (**b**) Scenarios 2 and 4; 95% significance. Letters over the bars are referred to the significance group according to the Tukey's range test.

The current sensitivity analysis shown in Figure 4 was performed based on the previous experimental drying tests conducted in pilot plant dryers. The uncertainties in the experimental results were taken into account, and a Monte Carlo simulation was performed based on these results. Moreover, due to the focus on climate change environmental issues, uncertainties related to cultivation and pulping stages were not considered, as these results are not expected to change during the analysis.

## 4. Conclusions

In this study, we conducted a comparative life cycle assessment (LCA) analysis of four scenarios of açaí powder production: vacuum drying, spray drying, and açaí powder production using both drying processes, with the added scenario in which the solid waste was used as fertilizer. In the stages of açaí cultivation, pulp extraction, and pulp conditioning, we found that the cultivation and drying processes, specifically the energy consumed in the drying process, had the most substantial environmental impact.

For different drying methods and scenarios, the carbon footprint of açaí powder production ranged from 7.93 to 8.26 kg of $CO_2$ eq. These scenarios suggest that energy consumption, particularly electricity, contributes significantly to the carbon footprint, with vacuum drying requiring more energy than spray drying.

Comparing different scenarios also revealed that the environmental impacts are generally similar, mostly due to the laboratory scale of the life cycle inventory data. However, scenarios that include processing residues as fertilizers do not significantly increase overall contamination, hinting at potential environmental and economic benefits of waste utilization in food production. Further research is required to determine the optimal method for utilizing açaí pulping residues as fertilizers, considering their nutrient content, availability, demand, and environmental impacts.

The results of the sensitivity analysis suggest that variations in drying temperature have minimal impact on the environmental footprint of the process. The energy needed to increase the air temperature from 120 °C to 140 °C or 200 °C was found to be relatively insignificant compared to the total energy consumed during the drying process. In similar fashion, the vacuum drying scenarios showed little variation in the environmental impact with changes in vacuum pressure.

In this study, we examine future dry açaí powder production systems in cultivation regions and identify several directions for optimization. These include reducing energy consumption during drying processes, utilizing processing residues as fertilizer, and scaling production to achieve greater efficiency. Currently, fertilizer is only applied during the nursery stage of açaí cultivation on jungle farms. While this study focuses solely on

environmental factors, the potential cost reduction resulting from the partial substitution of fertilizer may have significant economic implications, particularly when scaling production. Of the two drying processes studied, spray drying has the lowest carbon footprint. However, given the minimal reduction in energy consumption achieved by using lower temperature air for drying, it may be more advantageous to conduct drying tests with higher concentrations of solids in the açaí–maltodextrin stream. This could be achieved by increasing the encapsulant or pre-concentrating the fruit pulp. Utilizing processing residues as fertilizer and scaling production could further increase efficiency in drying.

Vacuum drying and spray drying technologies can extend the shelf life and lower the environmental impact of food powders derived from fruit surplus while reducing their carbon footprint and other environmental aspects. This work applied process modelling and life cycle analysis to conduct a preliminary evaluation of the conceptual design and the environmental assessment of a full-scale production process.

Future work should focus on (i) life cycle cost (LCC) analysis with the costs and benefits included, initial cost, operation and maintenance cost, collection and transport cost, and revenues from the sale of beneficial products and the revenue of disposal fees from government and (ii) scaling up the food powder production process that is technically, economically, and environmentally optimal by conducting sensory and consumer tests, determining the market form of the product, as well as addressing product safety and quality issues.

**Author Contributions:** N.S.-A.: Conceptualization, methodology, software, validation, investigation, data curation, writing—original draft; T.A.-P.: conceptualization, methodology, software, validation, investigation, data curation, writing—original draft; S.O.-C.: methodology, software, validation, data curation, formal analysis, writing—original draft; I.Á.-L.: validation, formal analysis, writing—review and editing; C.E.O.: conceptualization, formal analysis, resources, project administration, funding acquisition, writing—review and editing. All authors have read and agreed to the published version of the manuscript.

**Funding:** This work was supported by the "Ministerio de Ciencia, Tecnología e Innovación (MINCIEN-CIAS)", Colombia (contract number: FP44842-213–2018) and the "Universidad Nacional de Colombia Sede Manizales" (Hermes codes 55158 and 57755), Margaret McNamara Education Grants, and the European Union's H2020 research and innovation program under the Marie Sklodowska-Curie grant agreement No. 801586.

**Data Availability Statement:** All related data and methods are presented in this paper. Additional inquiries should be addressed to the corresponding author.

**Acknowledgments:** The authors express their gratitude to the the research program entitled "Reconstrucción del tejido social en zonas posconflicto en Colombia" SIGP code: 57579 with the project entitled "Competencias empresariales y de innovación para el desarrollo económico y la inclusión productiva de las regiones afectadas por el conflicto colombiano" SIGP code 58907, contract number: FP44842-213-2018; the project Evaluaciones técnica y de desempeño ambiental de productos de agregación de valor de cultivos de comunidades vulnerables de Chocó y Caldas (Hermes code 55158); the project Implementación de estrategias para incrementar la resiliencia al cambio climático y la agregación de valor a los productos derivados del cultivo del ají (Capsicum annuum) en el departamento de Caldas (Hermes code 57755); the Margaret McNamara Education Grants 2023; and the European Union's H2020 research and innovation program under the Marie Sklodowska-Curie grant agreement No 801586.

**Conflicts of Interest:** The authors declare no conflict of interest.

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
