# Peer review of "Environmental Life Cycle Analysis of Açaí (Euterpe oleracea) Powders Obtained via Two Drying Methods"

_processes, doi:10.3390/pr11082290_

Round 1

Reviewer 1 Report

1.The article is interesting to read and not having sufficient experimental evidence to support the discussion part, that made us too difficult to understand the concepts.

2. Excepting to include related articles carefully in the Introduction and discussion sections for Example see the article (The Use of Euterpe oleracea Mart. As a New Perspective for Disease Treatment and Prevention (mdpi.com).

3. Expecting some insight results and data for better clarifications

It would be more understandable if authors edit the minor English language.

Reviewer 2 Report

Please answer all questions and suggestion along the PDF manuscript

English quality most be imrpove

Reviewer 3 Report

The present work investigated the Environmental life cycle analysis of acai (Euterpe oleracea) powders obtained by different drying methods. This is an interesting idea to improve the high value use of Açaí by-products and save energy. However, several deficiencies in the manuscript require major revisions. The specific evaluation is as follows:

1. Can references be inserted into manuscript abstracts? Please confirm.

2. The introduction is rather lengthy but does not highlight the content and research focus that is closely related to the research objectives of the manuscript and needs to be further improved and refined in terms of writing quality.

3. It is suggested that the author supplement the introduction with a summary of why this research work should be carried out and what the specific significance is.

4. In the results and discussion, the authors may properly analyze the economic factors.

5. The authors should analyze in detail the advantages and disadvantages of each process and conclude to the best, with the best benefits and environmentally friendly implementation.

6. When discussing the unfavorable results, the authors should actively give the corresponding reasonable solutions.

7. Conclusion, the authors can supplement more aspects of the specific application of this study in the food industry and the outlook for future work.

Round 2

Reviewer 1 Report

1.The Entitled article explain about different types of acai powder drying for energy saving purpose. The author clearly explained the way of acai cultivation, that should be appreciable and could be understand the purpose of the research easily. 

2.Also, author put much effort to improve from the abstract, introduction and also discussion part. 

3. The article can be accepted at the present form, anyhow it would be better to cheek the small grammatical English.

1.The Entitled article explain about different types of acai powder drying for energy saving purpose. The author clearly explained the way of acai cultivation, that should be appreciable and could be understand the purpose of the research easily. 

2.Also, author put much effort to improve from the abstract, introduction and also discussion part. 

3. The article can be accepted at the present form, anyhow it would be better to cheek the small grammatical English.

Author Response

Thank you for taking the time to review both versions of our article. We appreciate your recognition of our efforts to improve the manuscript, which has been enhanced thanks to your important suggestions. In the revised manuscript we have included some adjustments in sepia font regarding the English language that colleagues who are fluent in English writing have suggested to us.

Reviewer 2 Report

Even if authors do not agree with suggestion, all question most be answered. Please verify all question are properly answered. Check the firts reviewed version

Moderate editing English language requiered

Author Response

We would like to express our gratitude and appreciation for the time you have spent reviewing the two versions of our article. We also would like to clarify that the questions and suggestions raised in the first round of review were answered throughout the PDF manuscript, as you requested. You can verify this by clicking on "Report Notes" (Round 1) on the journal platform. Virtually all your questions were answered and the article gained in quality thanks to these suggestions, for which we are very grateful. We apologize for two omissions: the modification of the title and the answer to the question on line 448. This was corrected in the new version of the manuscript. In order to avoid confusion, the responses given to your questions are transcribed below.

Regarding the English language, some minor modifications were made, which are highlighted in sepia letters in the corrected article.

---------------------------------------------------------------------------------

Responses to reviewer 2

Title: Environmental life cycle analysis of acai (Euterpe oleracea) powders- powders and residues ?- obtained by different drying methods ……different or two drying methods

 The life cycle analysis encompassed the entire cradle-to-gate process of producing dehydrated powders, using optimization strategies such as the use of waste. As a result of this emphasis, the title does not explicitly mention the inclusion of waste.

Indeed, there are only two drying methods evaluated, the modification of the title is made: Environmental life cycle analysis of acai (Euterpe oleracea) powders obtained by two  drying methods

Lines 20 to 24: The context of this sentence is not clear

The sentence was modified:  

“In this study, a cradle-to-gate Life Cycle Analysis (LCA) was conducted for the production of Açaí powder, following the methodology outlined in the ISO 14040 standard. The investigation focused on examining the impact of utilizing or not utilizing the residues generated during the pulp extraction step as fertilizers”

Lines 164-165: Two different raw material for the production of two different products using two different drying methods: spray drying and vacuum drying  ?

The four specified routes correspond to two types of drying methods (vacuum drying and spray drying) and the use or non-use of their residues. The main raw material was only one: the pulp of the acai fruit.

Line 169 (System boundaries): I am not sure if the words Boundaries (limits) and scenarios (possible situation) are the best technical words to describe acai and powder production syatem

The term ‘boundaries’ is derived from the ISO14040 and 14044 standards, which govern the execution of LCA. The term ‘scenario’ is commonly used in LCA studies to describe potential future situations, introducing temporal dimensions into the analysis

Figure 1: where scenario 3 y 4 are located, I can see Pulp conditioning only

In response to the comment, Figure 1 has been changed

Table 1: Word Value: what this means? units?

The term value means the quantities of compounds  required for  activities. The units of these values are the next column (i.e. kg, m3). It's not currency, as mentioned the header ‘Value’ means the quantity of each compound that you require to build the equipment. o prevent ambiguity, the term ‘Value’ has been replaced with ‘amount’.

Table 1, footnote: Please clarify the relationship of figure 1 and table 1. It is difficult to understand.

To clarify this relationship, the phrase above table 1 was rewritten: ‘The inputs necessary for the agronomic phase, including seed and plant production as depicted in Figure 1, for one hectare up to the initial harvest of the crop are detailed in Table 2’.

Lines 191-194: The germination rate- efficiency?- of the seeds in the nursery stage is 95%- % is not unit of rate, please explain-. Therefore, to obtain 625 trees/ha , an excess of seeds (656 seeds-656 are the execess needed? or 5 %?-) was required.

Thank you for your observations. The appropriate term is 'germination efficiency'. The manuscript has been updated to reflect this change: ‘The germination efficiency of seeds during the nursery phase is 95%. Consequently, to achieve a density of 625 trees per hectare, an additional quantity of seeds (656 seeds) is necessary ‘.

Line 208: intented or suggested?

This study was conducted at an industrial facility with its own acai crops, seeking to implement drying technology. Thus, the term 'suggested' is more appropriate. The necessary modification has been made.

Line 231: new product or subproduct?

In this instance, it is a subproduct. The necessary modification has been made.

Line 237: encapsulated powder? please explain

In both drying methods, the acai pulp was combined with maltodextrin (an encapsulant) prior to dehydration to preserve its nutritional properties, including polyphenolic compounds and antioxidant activity. This process is detailed in the Materials and Methods section.

Line 271: °Brix is a measure of soluble solid content or sweetness ?

°Brix represents soluble solids. As detailed in the Materials and Methods section, the pulp, which had a moisture content of 89.59%, was brought to 30°Brix using maltodextrin.

Line 275: what is draying yield?

Drying yield is calculated by dividing the mass of the powder produced by the spray or vacuum dryer by the mass of solids in the feed, and then multiplying by 100.

Lines 290-291: this sentence is not clear

The sentence was modified: 'The final moisture content represents the residual moisture present in the dried Açaí powder encapsulates. It was observed that the powder yield in scenario 1 was greater than that of scenario 2'.

Figure 2: How determine each parameter to construct the figure

The figures were constructed using data obtained from the SimaPro software, employing the ReCiPe Midpoint method. This method generates 16 categories, from which the most representative were selected for graphical representation.

Lines 325-327: This result is attribute but not measured

While it is widely acknowledged that contamination of primary organic nutrient sources with pathogenic microorganisms is a significant concern, this was not measured in the present study.

Lines 336-338: Confused argument

Figure 3 presents the life cycle impact evaluation of producing 1 kg of Açaí powder for the analyzed scenarios, using the ReCiPe Midpoint method (hierarchical version H) in SimaPro software. The figure is divided into four panels, each representing a different scenario: (a) Scenario 1 - Vacuum drying without waste use, (b) Scenario 2 - Spray drying without waste use, (c) Scenario 3 - Vacuum drying with waste use, and (d) Scenario 4 - Spray drying without waste use.

Lines 342-347: Please redifine each category

These definitions are widely known. However, there is a brief definition of them below, which however is not included in the manuscript: Climate Change (CC): Refers to long-term changes in the Earth's climate, including temperature, precipitation, and wind patterns, primarily as a result of human activities such as the burning of fossil fuels and deforestation.

Terrestrial Acidification (TA): Refers to the process by which acidic substances are deposited on the Earth's surface, leading to a decrease in soil and water pH, which can harm plants and wildlife.

Human Toxicity (HT): Refers to the potential harm to human health caused by exposure to toxic substances, including chemicals and pollutants.

Terrestrial Ecotoxicity (TET): Refers to the potential harm to terrestrial ecosystems caused by exposure to toxic substances, including chemicals and pollutants.

Freshwater Ecotoxicity (FET): Refers to the potential harm to freshwater ecosystems caused by exposure to toxic substances, including chemicals and pollutants.

Ionizing Radiation (IR): Refers to radiation with enough energy to remove tightly bound electrons from atoms or molecules, thus ionizing them. This can cause harm to living organisms by damaging DNA and other cellular structures.

Agricultural Land Occupation (ALO): Refers to the use of land for agricultural purposes, including crop production and livestock grazing.

Water Depletion (WD): Refers to the unsustainable use of water resources, leading to a decrease in the availability of freshwater for human and ecological needs.

Metal Depletion (MD): Refers to the unsustainable extraction and use of metal resources, leading to a decrease in their availability for future generations.

Fossil Depletion (FD): Refers to the unsustainable extraction and use of fossil fuels, leading to a decrease in their availability for future generations.

 (1) Global environmental impacts: data sources and ... - Springer. https://link.springer.com/article/10.1007/s11367-019-01604-y.

(2) Characterisation factors of the ILCD Recommended Life Cycle Impact .... https://eplca.jrc.ec.europa.eu/uploads/LCIA-characterization-factors-of-the-ILCD.pdf.

(3) Table 1 | Quantitative Methods for Life Cycle Assessment ... - Springer. https://link.springer.com/chapter/10.1007/978-3-319-68177-1_12/tables/1.

Line 435: sensivity analyses should be in material and methods

In accordance with the reviewer’s suggestion, the methodology employed in the sensitivity analysis has been incorporated into the Materials and Methods section.

Line 448: where are the drying processes curves to calvulate energy comsumption

The mass and energy balances were the input data used for the simulation of the drying processes that was carried out using the Superpro Designer software (Intelligen, Inc., USA)

Line 466 (conclusions): Should rewrite this section according with suggestions a long of manuscript

In response to the reviewer's suggestion, the Conclusions section has been completely revised. Please refer to the updated version of the manuscript.

Reviewer 3 Report

The authors have made more complete revisions based on the reviewers' comments and agree with the acceptance of the manuscript.

Author Response

We would like to express our gratitude and appreciation for the time you took to review both versions of our article. Thanks to your valuable recommendations, the article significantly improved in quality, for which we are very grateful.
